

# Elevated atmospheric mercury concentrations at the Russian polar station Amderma during Icelandic volcanoes' eruptions

Fidel Pankratov [a,*], Alexander Mahura [b], Tuukka Petäjä [b], Valentin Popov [c], Vladimir Masloboev [a]

[a] Institute of Northern Environmental Problems, Kola Science Center, Russian Academy of Sciences, Fersman Str. 14A, Apatity, 184200, Russia.
[b] Institute for Atmospheric and Earth System Research (INAR)/Physics, Faculty of Science, University of Helsinki (UHEL), P.O. Box 64, Helsinki, FI-00560, Finland
[c] Research and Production Association "Typhoon" of Roshydromet, Obninsk, Russia

*Correspondence to:* Fidel Pankratov (*fidel_ru@mail.ru*)

Abstract. We estimate the long-range atmospheric transport of elemental mercury in the Northern Hemisphere and present new data for volcanic eruptions in Iceland. At the polar station Amderma (Russia) of long-term observations of elemental mercury concentration (2009-2010), a change in the dynamics was recorded. For seasonal variability at the period from 2001-2009 negative trend (-0.66 ng/month) was fixed. However, the analysis of the last three years of measurement (2010-2012) showed the greatest positive trend (+0.97 ng/month). In April 2010 and the highest positive trend was observed (+0.24 ng m$^{-3}$), for the first time for the whole (2001-2013). At the same time, high concentrations of gaseous elemental mercury in the range from 1.81 to 2.58 ng m$^{-3}$ in April-June 2010 and from 1.81 to 3.3 ng m$^{-3}$ in May-June 2011 in contrast to the typical concentrations of 1.5 ng m$^{-3}$. During the period of 2010 and 2011 intensified volcanoes in Iceland and consequently volcanic eruptions in Iceland were considered the most probable cause of these increased concentrations. Until now, there have been no cases of recording a high concentration of mercury during the active eruption of the volcano, measured so far from the source of the eruption. In this way for the first time at the Amderma station in the Russian Arctic, high levels of elemental mercury were recorded as associated with the periods of active volcanoes Eyjafjallajokull (in 2010) and Grimsvotn (in 2011). The inverse trajectories calculated for a vertical profile covering a height of 500 and 3000 m above sea level the time level with high mercury concentrations confirmed that this was due to atmospheric transport from the northwest and was associated with the active Eyjafjallaicull and Grimsvotn volcanoes. Therefore, it can be assumed that these active volcanoes are the main sources of increased mercury concentrations in the northern hemisphere as a result of atmospheric transport of volcanic clouds to the monitoring point in the Russian Arctic.

## 1. Introduction

Many studies of global volcanogenic Hg emissions show that, in the main, the arrival of atmospheric mercury can be estimated relative to other compounds that are formed during degassing and eruption. During a volcanic eruption about 45 different trace elements from the Earth's crust can enter the atmosphere. However, there is no yet direct correlation between the amounts of the substance emitted into the atmosphere during an eruption and the number of reported cases of volcanic eruptions (Mambo et al., 2001). Volcanoes are considered to be the main natural sources of mercury entered into the atmosphere. The volcanogenic Hg flux from passively degassing volcanoes is about 30 Mg Hg yr$^{-1}$. The flux from erupting volcanoes is much larger; we estimate it at about 800 Mg yr$^{-1}$. Geothermal sources contribute to the atmosphere roughly 60 Mg yr$^{-1}$ Hg (Varekamp et al. 1986). Measurements of Hg speciation at the crater edge volcanoes Etna and Masaya have shown that over 90 % of the Hg was present in its elemental form. The particulate and reactive gaseous fraction comprised 1-5 % and ~1 % respectively of the total Hg (M. Li Witt et al. 2010). It should be noted also that on 13$^{th}$ Apr and during 17-21 Apr 2010 (Flentje at al., 2010) the elevated concentrations of particulate fractions (with particle size > 3 nm) were recorded at the global atmospheric monitoring stations (Zugspitze / Hohenpeissenberg - Germany, 2650 m above sea level, asl).

Hg levels in volcanic fumaroles in combination with sulfur analyses and published SO$_2$ flux data, allowed to estimate the global Hg flux. Usually, emissions of an element during the period of the eruption can be assessed based on a ratio of concentrations



between investigated element and sulfur dioxide $SO_2$ (Stoiber et al., 1987; Andres et al., 1998; Halmer et al., 2002). For active volcanoes the $Hg/SO_2$ ratios of $10^{-4}$, $10^{-4} – 10^{-6}$ and $10^{-6} – 10^{-7}$ for ash rich plumes in the active period (Ferrara et al., 2000; Nriagu and Becker, 2003; Pyle and Mather, 2003; Bagnato et al., 2009a). Usually, emissions of an element during the period of the eruption can be assessed based on a ratio of concentrations between investigated element and sulfur dioxide $SO_2$ (Stoiber

et al., 1987; Andres et al., 1998; Halmer et al., 2002).

A lot of questions about the volcanic origin of other elements are still remained (Hinkley et al., 1997), although there are almost no questions about the volcanic origin of mercury's (Nriagu et al., 2003; Pyle et al., 2003). However, both before the eruption of the volcano and at the end of the active phase of the eruption, through the cracks in the Earth's crust an active process of degassing (escape of large amounts of gases at the high temperature on the surface) is occurred. In particular, the mercury

concentration measured near the mud volcanoes on the Kamchatka Peninsula (Russia) is amounted within a range from 5-$7.5x10^{-5}$ to $2x10^{-3}$ g $l^{-1}$ of the condensate (Ozerova et al., 1988). Research conducted from 1991 to 1995 in the area of the Etna volcano showed that during the eruption the atmospheric concentrations of elements such as Bi, Cu, Cd, Sn and Zn had increased (Gauthier et al., 1998). Collected samples of magma showed that the concentrations of Hg and Bi are approximately equal. Hence, the ratio of these elements can be used for forecasting. Taking into account that the ratio of $Bi/SO_2$ concentrations

is in a range from $10^{-6}$ to $10^{-4}$ it can be assumed that the ratio of $Hg/SO_2$ will be also in the same range. Annual contribution of Bi is about 37 t year$^{-1}$ during active phases of eruptions, and it is about 5 t year$^{-1}$ during degassing (Gauthier et al., 1998). Main discussions about emissions during eruptions are focused on amounts of emissions that are produced by each particular volcano (Mather et al., 2003), and in which ratio the mercury in a gas phase with respect to mercury deposited on aerosol particles in the same volume is observed.

A usage of data retrieved from natural deposition of mercury in peat bogs or ice cores samples (Roos-Barraclough et al., 2002) or measurements of mercury concentration in a gas phase during volcanic eruption (Temme et al., 2003) can provide more accurate information about mercury flux into the atmosphere. (Temme et al. 2003) to test this assumption the modeling calculations were carried out. The backward trajectories were calculated for selected events with elevated mercury concentrations registered in the surface layer of the atmosphere.

The aim of this study is to provide an analysis of long-term gas-phase mercury concentrations in the Arctic observation site, Amderma, based on the data set from 2001 to 2013. We explore the seasonal cycle and trends of mercury concentrations and identify two emission sources for the atmospheric mercury, namely Eyjafjallajökull and Grimsvötn volcanic eruptions that influence the mercury concentrations during 2010 and 2011.

## 2. Methodology

### 2. 1. Long-term measurements of atmospheric mercury

Since June 2001 the long-term monitoring of the gaseous elemental mercury, GEM (hereafter, mercury) in the surface atmospheric layer was carried out near the Amderma settlement (69,72°N; 61,62°E; Yugor Peninsula, Russia) (Pankratov et al., 2008). It is located on the shore of the Kara Sea and close to the Arctic border between Europe and Asia (Fig 1).

To conduct long-term continuous measurements of mercury concentration in the Arctic atmosphere, the Tekran Instruments

Corporation (Toronto, Canada, http://www.tekran.com) developed the cold vapor atomic fluorescence spectrometer. This device is a mercury vapor analyzer «Tekran 2537A» capable of measurements at pg level. This analyzer was chosen as the primary instrument used for continuous measurements of mercury at all polar stations, including Amderma (Steffen, et al., 2005). Since 2005 measurements have been carried out with a time interval of 30 minutes. The monitoring has demonstrated that the background concentrations of elemental mercury in the surface layer in the Russian Arctic ($1.5 \pm 0.4$ ng m$^{-3}$) are similar

to the global background - $1.5 – 1.7$ ng m$^{-3}$ - in the Northern Hemisphere (Steffen, et al., 2008). The atmospheric mercury depletion events (AMDEs) were observed at Amderma each year too. During the depletion of atmospheric mercury, the mean mercury concentrations usually decrease below 1 ng m$^{-3}$ and with significant variability (Konoplev et al., 2005).



The polar station of Amderma as a monitoring point was chosen to assess the flux of mercury and POPs to the ecosystem of the Russian Arctic (Pankratov et al., 2010). A long-term (2001-2013) time series of obtained atmospheric mercury concentrations are shown in Fig. 2.

### 2.2. HYSPLIT trajectory modelling

Backward trajectory modeling is a widely used tool to assess the potential of atmospheric transport from different geographical locations where natural or anthropogenic sources of mercury may be present. Determination of air parcels spatial positions in the atmosphere during such movements may allow the identification of potential paths and regions where pollution can be found and how it is related to potential sources (Mahura at al., 2009). The accuracy of calculation of the trajectory is generally of the order of 20% of the travel distance, although in some cases the ratio may be even higher (Stohl et al., 1998). In our study,

the National Oceanic and Atmospheric Administration (NOAA) on-line transport and dispersion Hybrid Single-Particle Lagrangian Integrated Trajectory model (HYSPLIT) v 4.5 model available in an interactive mode was used (http://www.arl.noaa.gov/ready/open/hysplit4.html; Draxler et al., 2003; Rolph et al., 2003). To manage the process of trajectory modeling the meteorological gridded dataset (http://dss.ucar.edu/pub/reanalysis; NCEP / NCAR global Present – 1948) was used (Kalnay at al., 1996). Each trajectory was calculated for cases with elevated concentrations of mercury at the

polar station Amderma. For simplicity, only three trajectories arriving at the measurement site at levels 500, and 3000 m asl were calculated backward in time up to 120 hours (i.e. 5 days), while using the modeled vertical motion method. Due to the fact that there is a difference of 4 hours between the Amderma local standard time (LST) and Coordinated Universal Time (UTC) provided in dataset and model, all trajectories were calculated on a corrected time in order to match the corresponding measurements at LST. Then all trajectories were attributed in to sectors according to pathway of atmospheric transport.

### 3. Result and discussions

### 3.1. General features of mercury concentrations in Amderma

During the 15-year observation period since 2001 the analyzer "Tekran 2537A" was located at three points at different distances (from 8.9 to 0.2 km) from the coast of the Kara Sea. It is necessary to note that this experiment in practice of heavy metals monitoring in surface layer of the atmosphere at the Russian polar station Amderma was carried out for the first time

(Pankratov, 2015).

Analysis of the data measured at the Site № 1 showed that during 2001–2004 the average mercury concentration was 1.65±1.91 ng m$^{-3}$ (the maximum value was 75.5 ng m$^{-3}$ and the minimum value was 0.1 ng m$^{-3}$, which is the detection limit of the Tekran Instrument) (see Fig. 2, A–C). The results obtained suggest that Site № 1 located on the border of the region (about 9 km from the coast of the Kara Sea (see Fig. 5, A–C)), where mercury depletion events occur less intensively compared to the points near

the coastal zone of the Arctic seas (monitoring Sites № 2 and № 3 (see Fig. 2, C–F)). To identify cases where for a long time (at least 2 hours) in the surface layer of the atmosphere higher concentrations of mercury (> 1.81 ng m$^{-3}$) were recorded the term "Atmospheric Mercury Enhancement Events" (AMEEs) was used by analogy with AMDEs. The long-term series of atmospheric mercury concentration for the entire period of research is shown on Fig. 2. For Site № 2 (2005–2010) the average mercury concentration was 1.48±0.42 ng m$^{-3}$ and the maximum one was 14.53 ng m$^{-3}$. For this Site a decreasing trend of

mercury concentration was observed (see Fig. 2, C–D). In June 2010, the analyzer was installed at the point № 3 (about 200 meters from the coastline of the Kara Sea). Analysis of the data showed that from June 2010 to October 2013 the average GEM concentration was 1.38±0.84 ng m$^{-3}$, and the maximum one was 94.35 ng m$^{-3}$. During this period of time the average mercury concentration was decreasing (Fig. 2, E–F).

### 3.2. Seasonal variability of mercury

Analysis of long-term monitoring data showed that during 2001-2013 the mean concentration of atmospheric mercury was 1.5±0.4 ng m$^{-3}$. Overall, there is a decreasing tendency in the ambient air (Fig. 3). However, during 2010-2012, an increase of



atmospheric mercury concentration and its significant variability were observed, with average concentrations of 1.67±0.3 ng m$^{-3}$ and 1.32±0.3 ng m$^{-3}$ in 2001 and 2009, respectively.

Results of long-term monitoring analyses have underlined, that throughout 12 years of observations there were events of temporary decrease of mercury concentration (< 1.0 ng m$^{-3}$) in the surface layer of the atmosphere (Schroeder at al., 1998) in

a frequent manner. Such periods, called "atmospheric mercury depletion events" (AMDE) are presented in Fig. 2. These events occur every year, and mostly from late March to early Jun. The smallest variability of mercury concentration during AMDEs in the atmosphere has been registered mainly in winter.

As apparent from in 2010 (Fig. 3) the dynamics of mercury behavior in ambient air at the monitoring station Amderma was not typical compared with previous years. Elevated mercury levels were recorded in spring and summer, and one of the reasons

for this behavior could be assigned to long-range atmospheric transport from Iceland during the eruption of the Eyjafjallajökull volcano. We explore this connection in more detail. The estimate of atmospheric mercury from 13 Apr till 12 May 2010 demonstrated that the concentrations were significantly higher (1.81 - 2.75 ng m$^{-3}$) than the average long-term values (1.5 ± 0.4 ng m$^{-3}$) (Steffen, et al., 2005) characteristic for the Northern Hemisphere. The analysis of calculated pathways of air masses transport (e.g. trajectories) from Iceland suggests that in the middle and second half of April 2010 the measurement station

Amderma was in an area affected by the volcanic emissions, unlike other global stations of mercury monitoring (Alert, Canada and Ny-Ålesund, Norway). In addition, a similar behavior was also observed in 2011 during eruption of the other Icelandic volcano - Grimsvötn (Fig. 1).

For the overall observation period of measurements a negative trend (-0.35 ng m$^{-3}$) was obtained (Fig. 3,b). However, for the last 3 years a significant positive dynamics and its high variability in surface layer were identified (Fig. 3,c) (on average,

1.43±0.4 ng m$^{-3}$ and 1.55±0.7 ng m$^{-3}$ in 2010 and 2012, respectively). For this period, a positive trend (+0.12 ng m$^{-3}$) was registered. Note that such a significant increase in the concentration of mercury at the Amderma station was registered for the first time, and moreover, such behavior is not typical (if the long-term trend in the reduction of the mercury concentration in the atmosphere is considered). This fact can be explained if we take into account that in 2010 there was a volcanic eruption of Eyjafjallajökull, and in 2011 - of Grímsvötn (both in Iceland), and then the volcanic cloud passed over the area where the

monitoring of mercury was conducted. We show that the registered increase of mercury concentration at the Amderma station is a result of the regional long-range atmospheric transport of volcanic cloud (consisting of gases and aerosols) in the polar regions of the Russian Arctic. The arguments are listed in the following paragraphs.

First, in general, during Mar-May the mercury concentration was observed to decrease (-0.66 ng/period, Fig. 4). However, during the cases with volcanic activity, the corresponding trend is the opposite (Fig. 4b,c). At the same time there is

a general trend towards higher concentrations in the springtime, and from 2005 until 2008 such dynamics was observed. Between Apr and May 2010 (Fig.4,b) there was a large variability of concentrations ±0.52 ng m$^{-3}$. Analysis of the last three years (2010-2012) showed that in Apr the largest positive trend (+0.97 ng/month) was observed in 2010 (note similar dynamics was also in 2005); and the largest negative trend (-0.88 ng) was identified in 2003.

Second, from May until Jun 2011 an increase of concentrations was recorded. The positive trend for this time period was +1

ng/period (being the highest for the multi-year period of observations) (Fig.4,c) with a large variability of ±0.53 ng m$^{-3}$. In contrast, the largest negative trend (-0.1 ng/period) was observed in 2006.

For 2010 and 2011, analysis of seasonal time series revealed the existence of the third peak of high atmospheric concentrations of Hg in the spring-summer periods. The ratio of the mercury concentration of the second and third peaks in the spring-summer period was 0.93 for to 2010 and 0.94 to 2011. As seen in Figure 4, d, the third additional peak of high concentrations (marked

in brown) was observed before the main peak of the spring maximum. In the previous (2001-2009) and in the subsequent (2012-2016) years, two peaks of increased Hg concentration were not recorded. As a rule, during the spring-summer period only one peak of elevated mercury concentration was observed.





The most likely explanation for the unusual behavior of atmospheric mercury is the impact of volcanic eruptions. In 2010 there was a volcanic eruption of Eyjafjallajökull, and in 2011 - of Grímsvötn (both in Iceland). Therefore, the volcanic cloud can be passed over the area of the monitoring Station Amderma. Moreover, the Eyjafjallajökull eruption coincided with AMDEs during the polar sunrise in spring 2010. The registered increase of mercury concentration at the Amderma station can be postulated as the result of the regional long-range atmospheric transport of volcanic cloud (consisting of gases and aerosols) in the polar regions of the Russian Arctic.

### 3.3. Volcanic activity during 2010-2011
### 3.3.1. Eyjafjallajökull

The elevation of the Eyjafjallajökull volcano is about 1670 meters asl. with a crater of about 4 km in diameter. On 20[th] March 2010, the first phase of eruption took place, and at that time relatively low amount of volcanic ash was emitted into the atmosphere (Karlsdóttir at al., 2011). During the second phase of activity, from 7 untill 14 April 2010, a large cloud of volcanic ash was emitted into the atmosphere up to 9.5 km. (Sigmundsson et al. 2010, Schumann et al. 2011). Later on, during 16-19 April 2010, according to the (Icelandic MetOffice; a single C-band Doppler weather radar located in Keflavik) the volcanic emissions of gases and ash reached altitude of about 8.5 km. Subsequently, the ash layer was observed in different heights around Europe (Ansmann et al. 2010, Flentje et al. 2010, Petäjä et al. 2012, Pappalardo et al. 2013)

A direct shortest distance between the source (Eyjafjallajökull volcano) and the measurement site (Amderma) is about 3370 km, as seen in Figure 1. There are no large sources of both anthropogenic and biogenic emissions of mercury into the atmosphere at these high latitudes and along atmospheric paths of volcanic cloud from Iceland toward the location (Yugor Peninsula, Russia) of the measurement site.

The mercury concentration in the surface layer for this period was more than 2 ng m$^{-3}$ (in particular, on 10[th] April 04 UTC - 2.21 ng m$^{-3}$, 13[th] Apr 10 UTC - 2.71 ng m$^{-3}$ and 14[th] April 2010 06 UTC - 2.07 ng m$^{-3}$). These concentrations are the highest for the active phase of the Eyjafjallajökull volcano eruption (Fig. 5a). At this time, the volcanic cloud spread mainly in the eastern directions. Then, the cloud moved in the north-eastern direction and passed over the Scandinavian Peninsula. According to the UK MetOffice during the same period the volcanic cloud passed over the station Amderma. This passage could be the reason for the elevated atmospheric mercury concentrations recorded at the measurement site. On 16[th] April 2010 (18 UTC) the main atmospheric transport was within the north-west direction and later turned to south-west (http://www.metoffice.gov.uk/aviation/vaac/_data/VAG_160616.png). During the second episode the elevated concentrations of mercury were also recorded (18 April 06 UTC - 2.54 ng m$^{-3}$; 20 April 09 UTC - 2.53 ng m$^{-3}$; 22 April 00 UTC - 2.25 ng m$^{-3}$; and 24 April 2010 09 UTC - 2.36 ng m$^{-3}$). During the third episode (6-12 May 2010), the concentrations were also high and were more than 1.8 ng m$^{-3}$ (e.g. 9 May 20 UTC - 1.82 ng m$^{-3}$; and 12 May 2010 14 UTC - 2.06 ng m$^{-3}$) (as seen in Fig. 5a).

### 3.3.2. Grímsvötn

The elevation of the Grímsvötn volcano is about 1725 m asl, and the length of the caldera is about 2 km. Grímsvötn eruption began on 21 May 2011 (17 UTC), and at 21 UTC the volcanic ash released had reached a height of about 20 km according to observations of the Icelandic Met Office. The plume was transported over Europe and Russia (Kerminen et al. 2011, Tesche et al. 2012, Moxnes et al. 2014).

From 23 till 25 May 2011 the volcanic ash aerosol fraction passed over the Yugor Peninsula and the northern territories of the Polar Urals within the boundary layer, and continued moving in north-eastern direction. Note, there are no local mercury sources (industrial facilities) in the area, and hence, the elevated concentrations could be a result of volcano eruption followed by atmospheric transport toward the studied region of Amderma.

The duration of the period with elevated concentrations is about 5 days from the moment of start of registration of elevated concentration on 25[th] May (01 UTC; 1.97 ng m$^{-3}$) and ending on 28[th] May 2011 (00 UTC; 1.87 ng m$^{-3}$). These values obtained





are considered to be elevated, given that an average concentration (calculated for the entire measurement period from 2001 to 2012) was about $1.5 \pm 0.4$ ng m$^{-3}$. In particular, the elevated concentrations were registered during several days (29 May 00 UTC; 1.78 ng m$^{-3}$; and 30 May 2011 15 UTC; 1.77 ng/m$^3$). Starting from 30$^{th}$ May 2011 and later, there were no emissions of volcanic ash at high altitude. The following days, from 4$^{th}$ June 2011, the registration of high concentrations of mercury is

mainly related to regional atmospheric transport of mercury (and mainly in gas phase). At the Amderma station, the elevated concentrations were registered at the later dates as well: 4, 5, 7, 11, 14, 18, and 20 June, and the highest concentration – on 23$^{th}$ June 2011 (see Fig. 5b).

It should be noted that period of June-July 2011 could be considered as a period with a frequent occurrence of the AMDEs events (Pankratov et al., 2010). However, the registration of high concentrations of mercury during the polar spring time shows

that it can be associated with receiving of large amounts of mercury from a strong source, which in our case could be a volcano in an active phase of eruption.

### 3.4. Trajectory analysis connecting the volcanic activity and observation

Analysis of the calculated backward trajectories showed that, indeed, the eruption of the volcano Eyjafjallajökull could influence the increase in the concentration of mercury in the surface layer of the atmosphere at the Amderma station. The

trajectories were calculated for the vertical profile covering heights of 500 and 3000 m asl and backward in time up to 120 hours. It should be noted that the backward trajectories were calculated for the time period when the volcano was in the active phase of the eruption. Trajectories were calculated for the three cases (13 and 18 April, and 1 June 2010) when the concentration of mercury was about 2 ng m$^{-3}$ (see Fig. 6a).

From April untill May 2010, according to our data, we can assume that the observed elevated levels of mercury at the polar

station Amderma are the result of long-range atmospheric transport of volcanic cloud of gaseous and particulate fractions of components formed during the active phase of eruption of Eyjafjallajökull (see Fig.3,b). From May till Jun 2011, the increase of the mercury concentration was also a consequence of global atmospherics transport of volcanic cloud following the eruption of Grímsvötn (see Fig.3,c). For these cases the calculated backward trajectories showed arrival of air masses from higher altitudes (between 1-6 km), and it is typical for global atmospheric transport. Similarly, backward trajectories were calculated

for the time period when there was eruption of the volcano Grímsvötn. Atmospheric transport for 25$^{th}$ May 2011 showed movement of air masses in the middle troposphere at altitudes between 1-5 km. Note, that in Jun 2011 the air flow in most cases differs in height profiles characteristic for hemispheric atmospheric transport in the middle layers of the troposphere (see Fig. 6b).

Based on calculated trajectories, for another three cases (25 May, and 5 and 23 Jun 2011), it can be confirmed that high

concentrations of mercury co-inside with atmospheric transport from the north-west and linked with the active volcano Grímsvötn. For the case of 23$^{rd}$ Jun (17:00 UTC) the atmospheric flow with mercury is in gas phase, moving above the Novaya Zemlya Archipelago, arrived to the Yugor Peninsula from the north-east direction (similar to the Met Office UK calculations). Based on this assumption, it can be assumed that volcanoes are the large natural sources of mercury as during the active phase of the eruption, and possibly also during degassing, when there is no intense emission of volcanic species. Considering that the

active volcanoes such as Eyjafjallajökull and Grímsvötn are main sources of mercury in the Northern Hemisphere, the intake of Hg in the Arctic ecosystems can be calculated using mercury concentration measurement data following the period of the eruption. Moreover, some upcoming works (Sönke et al., 2016) shows that a lot of mercury is transported to the Arctic via rivers in water soluble form.

### 4. Conclusions

The annual average concentration of mercury decreased from $1.67 \pm 0.3$ (2001) to $1.32 \pm 0.3$ ng m$^{-3}$ (2009) during the studied period and a negative trend was observed. For seasonal variability at the period from 2001-2009 negative trend (-0.66



ng/month) also was fixed. For the period (2009-2010), a highest positive trend (+0.97 ng m$^{-3}$) was registered. However, the analysis of the last three years of measurement (2010-2012) showed the greatest positive trend (+0.97 ng/month). From Apr to May, 2010 (1.81 - 2.71 ng m$^{-3}$) and during May-August 2011 (1.81 - 3.69 ng m$^{-3}$) mercury concentrations were significantly higher but, the average long-term values for the Northern Hemisphere in the Russian Arctic were about 1.5 ± 0.4 ng m$^{-3}$. In this study a detailed analysis of long-term measurement data containing high concentrations of atmospheric mercury in the surface layer linked with the period of the eruptions of the two volcanoes Eyjafjallajökull and Grímsvötn (Iceland) was performed. Analysis of long-term (2001-2016) observations of gaseous elementary mercury concentrations had underlined time periods with high concentrations in the surface layer at the polar station Amderma. However, during these time periods (the polar spring time) registration of low concentrations of the mercury in the atmosphere (the cases AMDEs) was also observed. Therefore, we can conclude that the presence of high concentrations of the mercury is not typical for this time of the year, and the intense emission of mercury in this Arctic region is determined by the hemispheric scale atmospheric transport. Backward trajectories were calculated for the time periods when elevated concentrations were observed. Trajectory calculations showed that the source of the elevated mercury in the polar region of the Russian Arctic supposed to be connected with the active phases of the volcanoes' eruptions in Iceland. Also, atmospheric transport modelling estimates (based on analysis of the London Meteorological Centre) showed that the cloud of volcanic ash for both cases of volcanic eruptions (Eyjafjallajökull and Grímsvötn) passed over the territory of the Yugor Peninsula. Therefore, of gaseous elemental mercury, deposited on aerosol particles could influence the increase of the concentration in a specified location of the measurements (e.g., Amderma station) and in the whole of the Arctic Circle of the Northern Hemisphere. The registered increase of mercury concentration at the Amderma station is the result of the regional long-range atmospheric transport of volcanic cloud (consisting of gases and aerosols) in the polar regions of the Russian Arctic. Hence, inflow of the mercury into the Arctic during the two Icelandic volcanic eruptions occurred basically due to mercury present in a gas phase as well as due to fractions of mercury deposited on aerosol particles.

Thus, the registration of elevated atmospheric mercury concentrations at the Amderma field station in 2010 and 2011 is an anomaly in terms of the dynamics of global atmospheric mercury pollution and is due to volcanic eruptions in Iceland and atmospheric transport peculiarities in this part of the Arctic. To confirm this conclusion, it is necessary to compare detailed data on the dynamics of atmospheric mercury content at other polar stations during this period of time.

## 5. Acknowledgements

The authors gratefully acknowledge my colleague Strelnikov I., Kozulin S., and Balandin V. for continuous maintaining the mercury analyzer for the period 2001-2010. Authors appreciate the invaluable assistance from personal working at the Russian polar station Amderma. The authors are thankful to Mikushin A. and Guskov V. for continuous maintaining the mercury analyzer, as well as to staff of the hydro meteorological service engaged in monitoring and technicians for their permanent support in performing measurements for the period 2010-2016. The authors gratefully acknowledge the NOAA Air Resources Laboratory (ARL) for the provision of the HYSPLIT transport and dispersion model and/or READY website (http://www.arl.noaa.gov/ready.html) used in this publication. Financial support for the monitoring program was provided by Environment Canada, Arctic Monitoring and Assessment Programme (AMAP) Secretariat and Russian Federal Service for Hydrometeorology and Environmental Monitoring (Roshydromet). For this study, the financial support was also partially provided by the Kola Science Center of the Russian Academy of Sciences and the Pan-Eurasian EXperiment programme. PEEX and the European Union's Horizon 2020 research and innovation programme under grant agreement No 689443 via project iCUPE (Integrative and Comprehensive Understanding on Polar Environments).

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





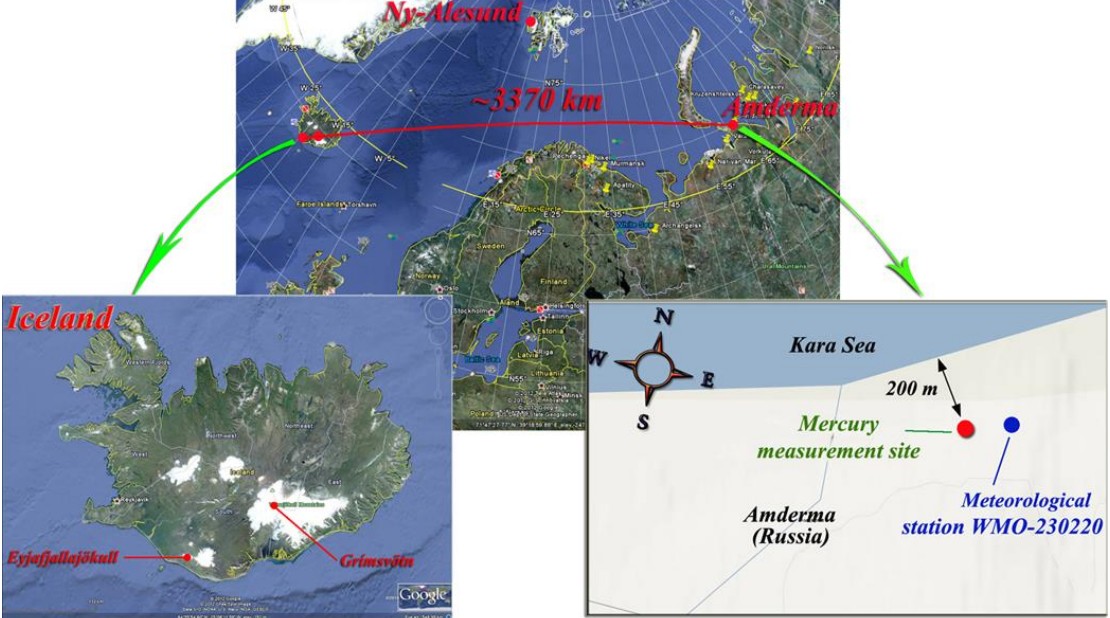

**Figure 1. Geographic layout of the Russian Arctic station (Amderma, Yugor Peninsula) for elemental mercury monitoring with respect to sources (volcanoes Eyjafjallajökull and Grímsvötn in Iceland) of mercury releases into the atmosphere.**





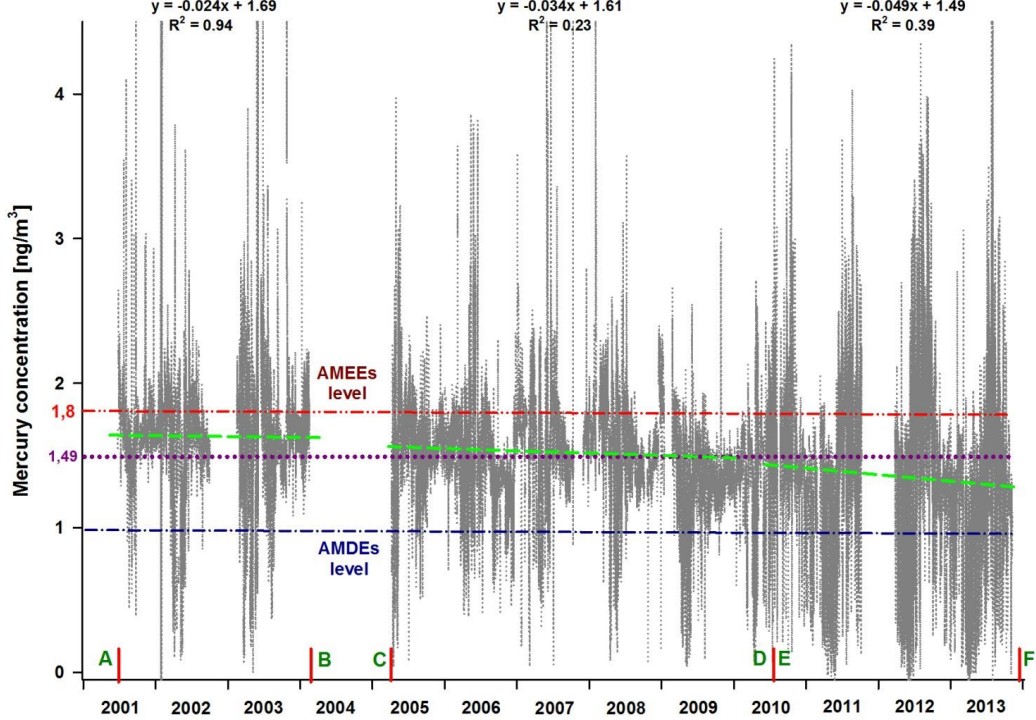

**Figure 2. The long-term series of atmospheric mercury concentration for the Polar Station "Amderma" for the period 2001–2004 (A, B), 2005–2010 (C, D), and 2010–2013 (E, F). For the period 2001–2013: mercury concentration above the red dash-dotted line with two points is the AMEEs area; green dashed line is the linear approximation of the average annual values; mercury concentration below the purple dotted line is the AMDEs area (Pankratov, 2015).**



Table 1. Average annual concentrations (ng m$^{-3}$) of total gaseous mercury in the Northern Hemisphere, (Reports of the Academy of Science (RAS), and Swedish Environmental Research Institute Ltd. (IVL)) (Pankratov, 2015).

| Location | Coordinates | 2001 | 2002 | 2003 | 2004 | 2005 | Source |
|---|---|---|---|---|---|---|---|
| Andoya, Norway | 69$^0$N, 16$^0$E | | | | 1.64 | | IVL, 2010 |
| Pallas, Finland | 68$^0$N, 24$^0$E | 1.41 | 1.48 | 1.57 | 1.49 | 1.63 | IVL, 2010 |
| Ny-Alessund, Norway | 79$^0$N, 12$^0$E | 1.61 | 1.63 | 1.63 | 1.52 | 1.61 | IVL, 2010 |
| Amderma, Russia | 69$^0$N, 61$^0$E | 1.65 | 1.73 | 1.71 | 1.52 | 1.54 | Pankratov F., 2013 |
| Northern Sea Route | 58$^0$N - 173$^0$E | 0.32 | | | | | RAS., vol. 322, 2002 |
| Kara Sea, Russia | | | 0.89 | | | | RAS., vol. 322, 2002 |
| Barents Sea, Russia | | 0.61 | | | | | RAS., vol. 322, 2002 |





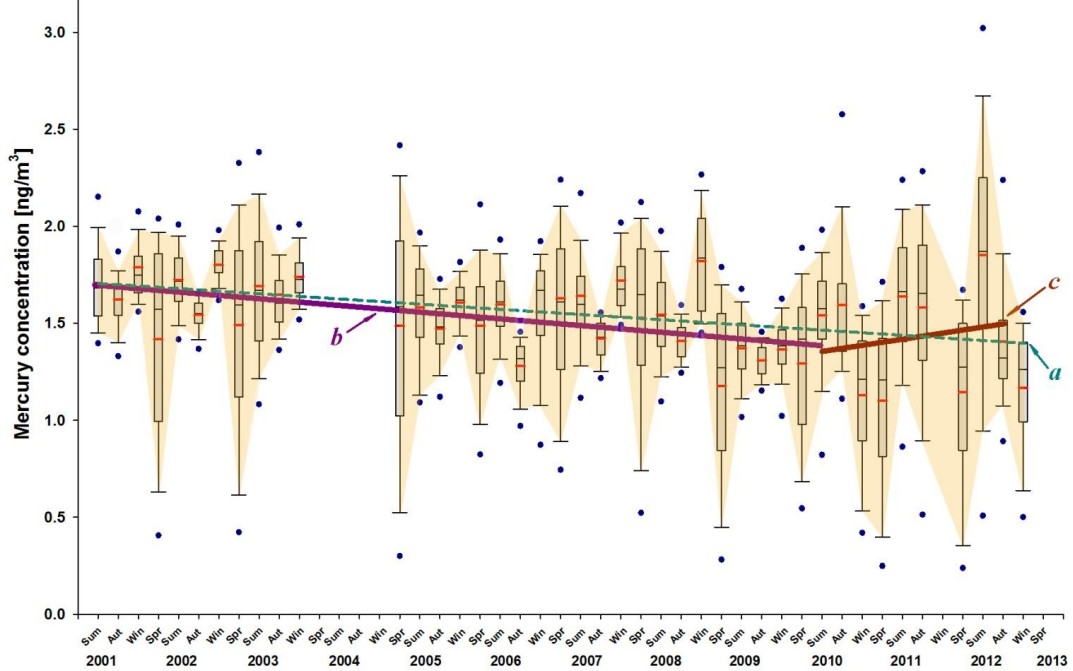

**Figure 3.** Seasonal variability of long-term (2001-2013) observations of elemental mercury concentration in the atmospheric surface layer at the Amderma station with linear trends for periods: (a) 2001-2013, (b) 2001-2009, and (c) 2010-2012. Comments: The bars indicate seasonally averaged data. The red line indicate the median, boxes depict quartile ranges and whispers 5 and 95 percentiles. The blue dots illustrate statistical outliers. The black line is the mean concentration for a given seasonally.

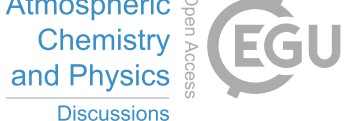



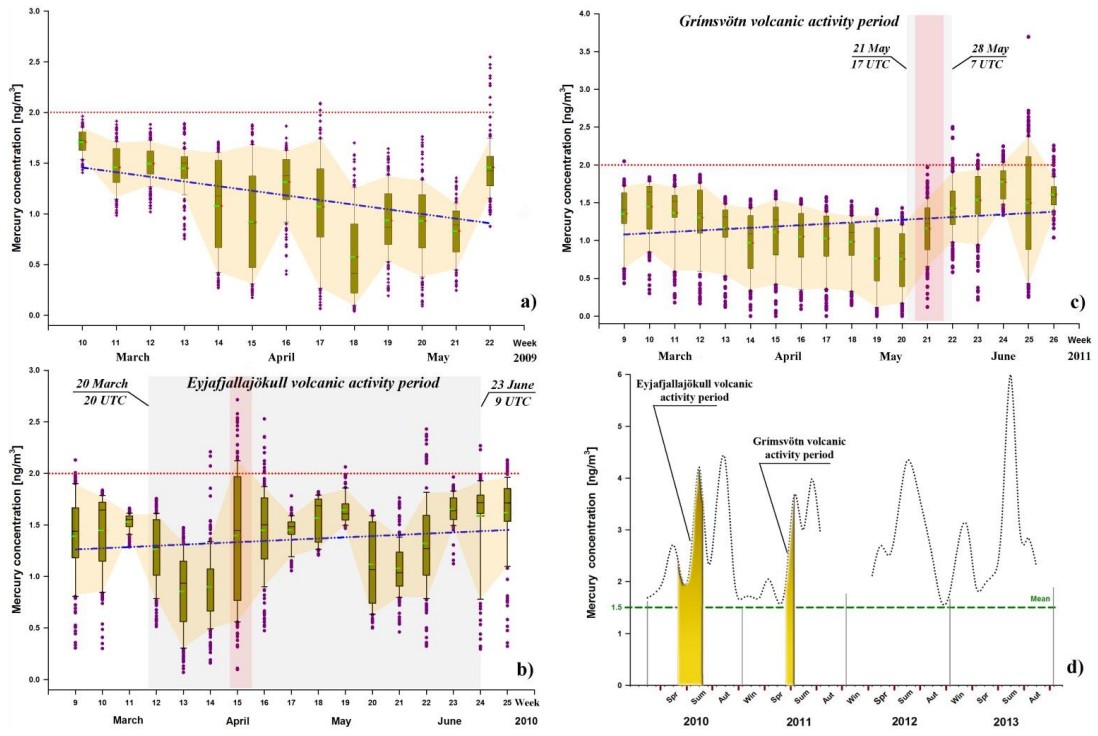

Figure 4. The behavior dynamics of elementary mercury concentrations: (a) for March-May 2009, (b) for March-June 2010, (c) for March-June 2011 during periods of the active phases of volcanic eruptions (b) Eyjafjallajökull and (c) Grímsvötn /gray rectangles/ and during periods of atmospheric transport of volcanic cloud passed over the Yugor Peninsula /red rectangles/; and (d) area high concentration during the volcanic eruption (2 peaks maximum / brown color).





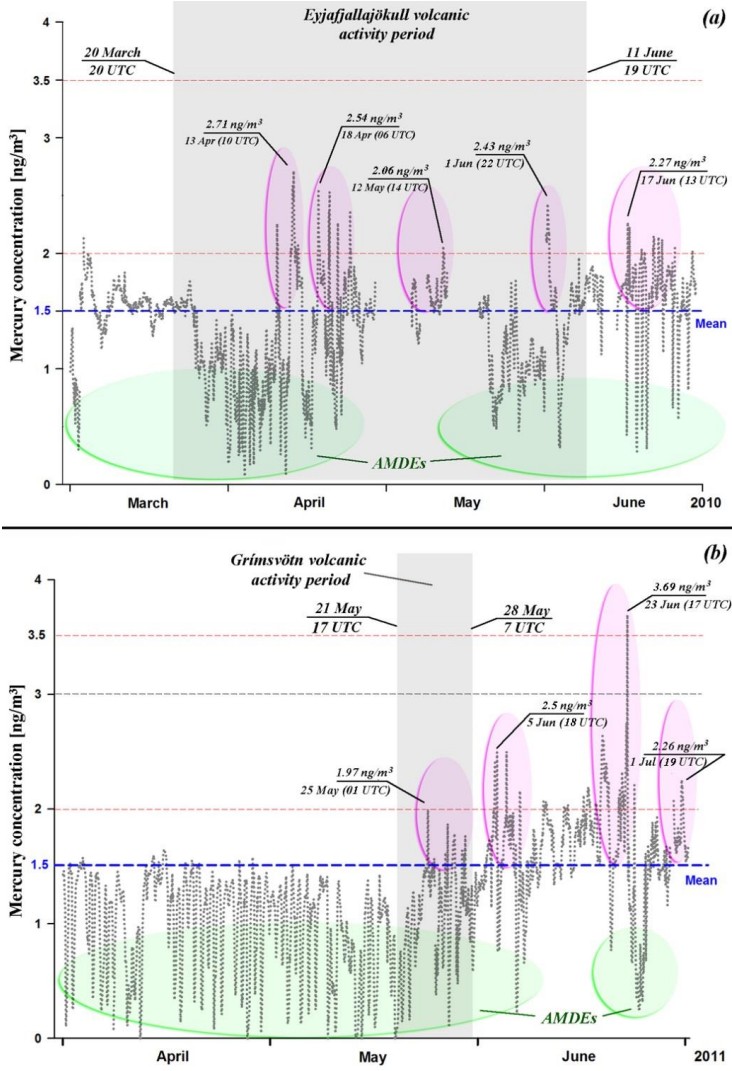

**Figure 5. Results of monitoring elemental mercury concentrations at the Russian station Amderma during the time of impact of the volcanoes Eyjafjallajökull and Grímsvötn: (a) March-May 2010 (b) April-June 2011. /The green ovals show the spring**
5 **atmospheric mercury depletion events (AMDEs). The red ovals indicate increased values of mercury concentrations resulting from air mass transport from the direction of erupting volcanoes in Iceland/.**





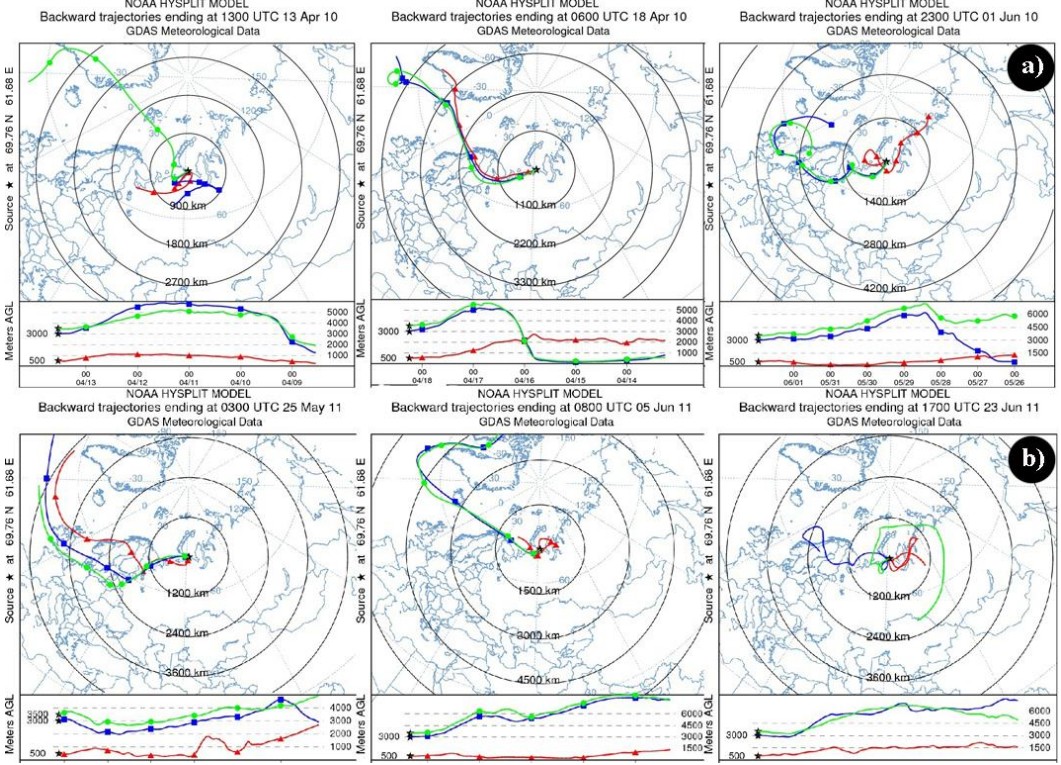

**Figure 6. Backward trajetories (calculated by the NOAA HYSPLIT model) arriving at the Amderma monitoring station on (a) 13, 18 April and 1 June 2010 and (b) 25 May and 5, 23 June 2011 and corresponding to eruprions of the (a) Eyjafjallajökull and (b) Grímsvötn volcanoes.**