# Peer review of "Atmos. Chem. Phys. Discuss., https://doi.org/10.5194/acp-2018-1228 Manuscript under review for journal Atmos. Chem. Phys. Discussion started: 21 December 2018"

_Atmospheric Chemistry and Physics, 2018_

## Referee Comment (RC1) · Anonymous Referee #1 · 26 Dec 2018

The authors present long-term mercury measurements at a remote site of Amderma on the coast of Kara Sea south of Novaya Zemlya in Russia. They observed events with unusually high concentrations in 2010 and 2011 and attribute them with help of backward trajectories rather convincingly to mercury released by volcanic eruptions in Iceland.

There are several problems with this paper. First, it is written more like a fiction than a scientific paper. Essential information such as description of the site, its climatology, instrument calibration, is missing. Second, beside a rather poor English and poor organisation of the manuscript it contains a number of errors (such as wrong units

for the discussed trends, faulty citations) and omissions (some crucial and more recent references are not mentioned, calculation of the trends is not defined). Third, I think that episodic transport of air masses with high mercury concentrations from volcanic activity in Iceland would be better discussed in event (and their frequency) terms or as an anomaly of seasonal variation, rather than in terms of trends whose discussion makes a substantial part of the paper. In summary, the paper does not meet the standard of a paper in Atmos. Chem. Phys. It could be published after a thorough and far reaching revision of its concept of presentation, figures and text. Some recommended revisions are listed below:

The references are generally not up to date – the most recent estimate of volcanic mercury source by Pirrone et al. (Atmos. Chem. Phys., 10, 5951-5964, 2010) is not mentioned. Pankratov's own paper in Russian Meteorology and Hydrology (Vol. 38, 405-413, 2013) with some experimental details on the Anderma station is not mentioned either. Conference abstracts should not be cited because they are difficult to come by.

Page 1, Abstract:

What does it mean "we estimate the long-range transport" – flux or what?

"New data for volcanic eruptions in Iceland" – source strength or what?

"A change in dynamics" – dynamics of what – transport, meteorology?

"For seasonal variability . . . a negative trend of.. ng/month was fixed" – a trend of seasonal variability or of concentrations? Fixed? Unit of ng/month?

The sentence in line 15 starting with "At the same time" is incomplete.

"The inverse trajectories. . ." backward trajectories are probably meant.

The last sentence reads either as if volcanic emissions were dominating source of atmospheric mercury in the northern hemisphere or as if Amderma station were repre-

sentative for the northern hemisphere or both. Both is wrong.

Page 1, Introduction:

Line 34: "we estimate it at about 800 Mg yr-1" – there is neither a reference nor an estimation presented in this paper.

Line 38: The citation of M. Li Witt, 2010 is a conference paper difficult to obtain. In addition, the initials of Ms Witt are wrong. Please cite M.L.I. Witt et al. (J. Geophys. Res. 113, B06203, doi:10.1029/2007JB005401, 2008; J. Volcanology Geotherm. Res. 178, 636-643, 2008).

Lines 38-40: It is not clear what elevated particle concentrations at Zugspitze have to do with mercury or other trace metals?

Line 41 and the last line of the same paragraph say the same, one of the sentences is redundant.

Page 2, lines 32-33: There are 4 citations of work by Pankratov of which three are conference abstracts difficult to obtain and thus useless to most readers. Please cite only Pankratov Thesis with the doi number which is accessible on internet. Why is the paper by Pankratov et al. (Russian Meteorology and Hydrology, Vol. 38, 405-413, 2013) not cited? It provides some valuable details about the Anderma measurements and an analysis of data until 2011.

Section 2.1: The site has to be described, possible mercury sources in the vicinity enumerated, and precautions to eliminate contaminated data delineated. There are no details about the air sampling (flow rate, position of inlet, tubing, its length and material), maintenance (exchange of gold traps, etc.) and calibration (frequency of calibration of Tekran response to mercury and of air flow rate) of the Tekran instrument. Please add details. A comparison with other sites in the northern hemisphere would be better placed in the discussion.

Page 3, line 15: Backward trajectory arrival altitude of 500 and 3000 m - is this altitude

justified? What is the orography and typical meteorology of the station (possible mercury sources in the surrounding, prevailing wind direction, velocity, height of inversion layer in different seasons)?

Section 3.1: In Fig. 1 the authors show the position of one of the three measurements sites – why not the position of other two? What was the position of site 1, 2 and 3, i.e. distance from the sea? By the way the measurements at the three sites were already compared in the Pankratov et al. 2013 paper. What about local contamination: has it been observed and eliminated from the data?

Section 3.2: This section is nominally about seasonal variation but it deals to a large extent with trends. Figure 3 is not helpful either. The authors should divide the discussion into a section about trends (if neccessary at all) and into a section about seasonal variation after detrending the data.

Page 4, lines 6-7: "The smallest variability of mercury concentrations during AMDEs. . ." – what does it mean? AMDEs mean high variability, the smallest variability would be an absence of AMDEs.

Page 4, line 8: "dynamics of mercury behaviour" – why "dynamics" and not simply "mercury behaviour"?

Page 4, lines 11-14: Because of the trend it would be more appropriate to compare the authors data in 2010 and 2011 with corresponding annual data from Ny Alesund and Alert stations. The data can be found in the literature or obtained from the operators of the stations. The term "significantly" should be accompanied by a significance test.

Page 4, lines 18-27: Trends have to have a unit of concentrations per time. The trends presented here in concentration units cannot thus be trends. Were the trends calculated from all data, from monthly averages or monthly medians? Fig. 2 shows that since about 2011 the frequency of the AMDEs and AMEEs is much higher than in preceding years without changing much the decreasing tendency. The discussed trend is

then likely to be caused by changing frequency and duration of depletion and enhancement events instead of the trend of background. I think that to discuss the 2011 and 2012 anomaly in terms of trend is thus wrong. It is also questionable whether 3 years are sufficient to calculate a trend.

Page 4, lines 28-33: Trend is given in ng/period – the unit is wrong. Which period? Short term changes are not trends!

Section 4: Trend units are wrong.

Section 4.3: In addition to backward trajectories, the long-range transport of volcanic plumes can be documented by remote sensing of SO2 by satellite. Details can be found e.g. in Heue et al. (ACP, 10, 4699-4713, 2010) who used the satellite SO2 sensing to follow the Kasatochi plume from Aleutian Islands in Pacific Ocean to Central Europe.

Page 7, line 7: Mercury measurements at Amderma seem to continue until 2016. By including later data, the anomaly of 2010 and 2011 could be perhaps illustrated better.

Figure 1: This figure could serve as a schematic illustration of the paper content (such as required by Environmental Science and Technology) but is unsuitable for a paper. The insert with the map of Island and its volcanoes can be found in every atlas and can thus be omitted. An insert with a map of the Anderma site (and the three sampling locations mentioned in the text) and the potential mercury sources in the surroundings would be more useful.

Figure 2: The data shown in this figure for 2002- 2004 are very different from those shown in Figure 2 of Pankratov et al. (2013) for the same period. Please explain the difference.

Figure 3: The decreasing trend in Figure 2 for 2010 – 2013 is now an increasing trend in Figure 3? This only shows that the discussion of volcanic influence in terms of trends is questionable at best.

Figure 4: The reader would appreciate the same scale on the x axis – why a) stops at the 22nd week when b) continues until 26th and c) until 25th week? The dotted line in d) is not explained – a running average or what?

Reference Konoplev et al. (2005) -the title and the page numbers are wrong.

---

## Referee Comment (RC2) · Anonymous Referee #2 · 7 Feb 2019

This paper presents some very interesting data and merits publication, just not in its current form.

In general, the paper needs English re-editing, the grammar and phrasing make it difficult to read at times. There are some spelling problems, e.g. page 1 line 24 a different spelling of the Icelandic volcano is used, or is it a different volcano (understandable seeing how hard Icelandic spelling is, but more care could be taken)?

Page 6 line 30, coincide not co-inside

A lot of the references are quite old, the thesis was published in 2015, but this paper is 2019, mercury science has moved on, the authors should carry out a new literature

search and include papers from 2015 onwards. Where conference papers have been cited, the authors should see if the conference paper lead to a real one, if not, the data should be disregarded, they can be safely eliminated from the references and replaced with a more modern reference.

This paper demonstrates that some of the peaks in the long-term trends of measurements could come from volcanic eruptions that occurred in 2010 and 2011 in Iceland. However, before this can be successfully shown to be true it would be useful to know where the other peaks come from so they can be eliminated as possibilities from this analysis. There are similar sized peaks in 2012 and 2013 that are not identified or discussed that probably aren't volcanic eruptions. What are their sources? They look very similar to the volcanic eruptions, are we sure that the cause isn't the same? A similar observation can be made for the data from 2009 back to 2001, there are frequent peaks of data above 4 ng/m3 but no description of their sources or causes, why are they so similar, and how can we be sure that the peaks in 2010-2011 are not caused by the same sources with volcanic plumes superimposed? The authors should discuss the whole data set and identify the other peaks, or they should not include them and talk about in more detail the volcanic data.

---

## Short Comment (SC1) · 14 Feb 2019

1. Essential information such as description of the site, its climatology, instrument calibration, is missing. Remarks partially accepted. A brief description of the site is sufficient for the purposes of this article. The authors believe that a too detailed description of climatology in the paper is excessive. The article provides the most significant information on the measurement modes. There indicated the accumulation period (30 minutes) and the desorption mode (thermal desorption). More information on calibration can be obtained from the link provided in the article on the Tekran website. 2. such as wrong units for the discussed trends Remarks accepted. Units

have been changed. 3. Third, I think that episodic transport of air masses with high mercury concentrations from volcanic activity in Iceland would be better discussed in event (and their frequency) terms or as an anomaly of seasonal variation, rather than in terms of trends whose discussion makes a substantial part of the paper. Remarks rejected: Episodic transport of air masses with high mercury concentrations from volcanic activity in Iceland are thoroughly described in section 3.2 "Episodes of volcanic eruptions" as an anomaly of seasonal variation. The calculation of the trend for periods of volcanic eruptions made it possible to estimate the degree of increase in the concentration of atmospheric mercury. This approach suggested that the increase in mercury concentrations in spring was associated with volcanic eruptions in Iceland. 4. the most recent estimate of volcanic mercury source by Pirrone et al. (Atmos. Chem. Phys., 10, 5951-5964, 2010) is not mentioned. The remark has been accepted. The reference "Pirrone et al. (Atmos. Chem. Phys., 10, 5951-5964, 2010)" will be added to the list of references. 5. Pankratov's own paper in Russian Meteorology and Hydrology (Vol. 38, 405-413, 2013) with some experimental details on the Amderma station is not mentioned either. The remark has been accepted. This reference "Russian Meteorology and Hydrology (Vol. 38, 405-413, 2013)" will be added to the list of references. 6. What does it mean "we estimate the long-range transport" – flux or what? Reply to comment. The term "long-range transport" means the transfer of a pollutant with air masses over long distances. 7. New data for volcanic eruptions in Iceland Reply to comment. For the first time, an increase in the concentration of elemental mercury in the surface layer of the atmosphere at a particular point in the Russian Arctic was associated with volcanic eruptions.. 8. A change in dynamics" – dynamics of what – transport, meteorology Reply to comment. In this case, we talk about the dynamics of atmospheric mercury for the period when the air masses transferred a significant amount of mercury over the measurement site of the monitoring from the south-west direction.

9. For seasonal variability . . . a negative trend of.. ng/month was fixed" – a trend of seasonal variability or of concentrations? Remarks accepted. The mercury concentration was expressed as ng per month.

10. The sentence in line 15 starting with "At the same time" is incomplete.

11. The inverse trajectories. . ." backward trajectories are probably meant. Remarks accepted.

12. The last sentence reads either as if volcanic emissions were dominating source of atmospheric mercury in the northern hemisphere or as if Amderma station were representative for the northern hemisphere or both. Both is wrong. Reply to comment. In this case, the assumption is confirmed that the maximum amount of mercury from volcanic eruptions supply to the environment in a short period of time. This statement is based on data from an ice core study. (Krabbenhoft, D.P. and Schuster, P.F., 2002 ,Glacial Ice Cores Reveal A Record of Natural and Anthropogenic Atmospheric Mercury Deposition for the Last 270 Years: 2002 U.S. Geological Survey Fact Sheet FS-051-02, p. 2.)

13. Introduction: Line 34: "we estimate it at about 800 Mg yr-1" – there is neither a reference nor an estimation presented in this paper. Remarks accepted. Link to article will be added

14. The citation of M. Li Witt, 2010 is a conference paper difficult to obtain. In addition, the initials of Ms Witt are wrong. Please cite M.L.I. Witt et al. (J. Geophys. Res. 113, B06203, doi:10.1029/2007JB005401, 2008; J. Volcanology Geotherm. Res. 178, 636-643, 2008). Remarks accepted. Link will be corrected.

15. Lines 38-40: It is not clear what elevated particle concentrations at Zugspitze have to do with mercury or other trace metals? Reply to comment. An example of registration in the atmosphere of elevated concentrations of other particles during the passage of a volcanic cloud above the site of the monitoring.

16. Line 41 and the last line of the same paragraph say the same, one of the sentences is redundant. Remarks accepted. The paragraph will be deleted. 17. Page 2, lines 32-

33: There are 4 citations of work by Pankratov of which three are conference abstracts difficult to obtain and thus useless to most readers. Please cite only Pankratov Thesis with the doi number which is accessible on internet. Remarks accepted. Link to article will be added.

18. Why is the paper by Pankratov et al. (Russian Meteorology and Hydrology, Vol. 38, 405-413, 2013) not cited? It provides some valuable details about the Amderma measurements and an analysis of data until 2011. Remarks accepted. Link to article will be added.

19. Section 2.1: The site has to be described, possible mercury sources in the vicinity enumerated, and precautions to eliminate contaminated data delineated. There are no details about the air sampling (flow rate, position of inlet, tubing, its length and material), maintenance (exchange of gold traps, etc.) and calibration (frequency of calibration of Tekran response to mercury and of air flow rate) of the Tekran instrument. Please add details. Remarks accepted. Technical details will be added.

20. Page 3, line 15: Backward trajectory arrival altitude of 500 and 3000 m - is this altitude justified? What is the orography and typical meteorology of the station (possible mercury sources in the surrounding, prevailing wind direction, velocity, height of inversion layer in different seasons)? Reply to comment. To calculate the Backward trajectory using different altitude. Including the altitude of 3000 meters is the height of the main transport in the atmosphere, the free troposphere.

21. Section 3.1: In Fig. 1 the authors show the position of one of the three measurements sites – why not the position of other two? What was the position of site 1, 2 and 3, i.e. distance from the sea? By the way the measurements at the three sites were already compared in the Pankratov et al. 2013 paper. What about local contamination: has it been observed and eliminated from the data? Reply to comment. The location of monitoring sites was described in the article Pankratov, F .: The Dynamics of Atmospheric Mercury in the Russian Arctic, Thesis, November 2015, DOI: 10.13140

[Figure]

/ RG.2.1.4255.1767. This information does not represent a value for the fact of registering elevated mercury concentrations at volcanic eruption in Iceland. Local sources of pollution do not significantly affect the resulting values of mercury concentration over the entire observation period.

22. Section 3.2: This section is nominally about seasonal variation but it deals to a large extent with trends. Figure 3 is not helpful either. The authors should divide the discussion into a section about trends (if necessary at all) and into a section about seasonal variation after detrending the data. Reply to comment. In this section, seasonal variations are treated as trends for certain time intervals.

23. Page 4, lines 6-7: "The smallest variability of mercury concentrations during AMDEs. . ." – what does it mean? AMDEs mean high variability, the smallest variability would be an absence of AMDEs. Remarks accepted. This sentence will be adjusted.

24. Page 4, line 8: "dynamics of mercury behaviour" – why "dynamics" and not simply "mercury behaviour"? Remarks accepted. This sentence will be adjusted.

25. Page 4, lines 11-14: Because of the trend it would be more appropriate to compare the authors data in 2010 and 2011 with corresponding annual data from Ny Alesund and Alert stations. The data can be found in the literature or obtained from the operators of the stations. The term "significantly" should be accompanied by a significance test. Reply to comment. If possible, I will request data from other stations.

26. Page 4, lines 18-27: Trends have to have a unit of concentrations per time. The trends presented here in concentration units cannot thus be trends. Were the trends calculated from all data, from monthly averages or monthly medians? Fig. 2 shows that since about 2011 the frequency of the AMDEs and AMEEs is much higher than in preceding years without changing much the decreasing tendency. Reply to comment. Observation period (2001-2010) of measurements a negative trend (-0.35 ng m-3), and calculated averages for all seasonal.

[Figure]

27. Then likely to be caused by changing frequency and duration of depletion and enhancement events instead of the trend of background. I think that to discuss the 2011 and 2012 anomaly in terms of trend is thus wrong. It is also questionable whether 3 years are sufficient to calculate a trend. Reply to comment. The construction of a trend for a three-year period is reasonable, since there are no time and quantitative limitation when analyzing any series of values.

28. Page 4, lines 28-33: Trend is given in ng/period – the unit is wrong. Which period? Short term changes are not trends! Reply to comment. -0.66 ng per period, Fig. 4, +0.97 ng per month, -0.88 ng per month. 29. Section 4: Trend units are wrong. Remarks accepted. Trend units will be corrected.

30. Page 7, line 7: Mercury measurements at Amderma seem to continue until 2016. By including later data, the anomaly of 2010 and 2011 could be perhaps illustrated better. Figure 1: This figure could serve as a schematic illustration of the paper content (such as required by Environmental Science and Technology) but is unsuitable for a paper. The insert with the map of Island and its volcanoes can be found in every atlas and can thus be omitted. An insert with a map of the Amderma site (and the three sampling locations mentioned in the text) and the potential mercury sources in the surroundings would be more useful. Reply to comment. The possibility of mapping the likely sources of mercury contamination will be considered. 31. Figure 2: The data shown in this figure for 2002- 2004 are very different from those shown in Figure 2 of Pankratov et al. (2013) for the same period. Please explain the difference. Reply to comment. The original data is not changed, a variety of mathematical techniques used to construct graphs. 32. Figure 3: The decreasing trend in Figure 2 for 2010 – 2013 is now an increasing trend in Figure 3? This only shows that the discussion of volcanic influence in terms of trends is questionable at best. Reply to comment. For the period 2010-2013 (Fig. 2) a downward trend was calculated. The trend to increase (Fig. 3) is calculated for the period 2010-2012. 33. Figure 4: The reader would appreciate the same scale on the x axis – why a) stops at the 22nd week when b) continues until 26th

and c) until 25th week? The dotted line in d) is not explained – a running average or what? Reference Konoplev et al. (2005) -the title and the page numbers are wrong. Reply to comment. For the graphs presented in the figure information on the number of weeks is not significant. The link to the article will be corrected.
* * *

---

## Short Comment (SC2) · 14 Feb 2019

1. In general, the paper needs English re-editing... There are some spelling problems, e.g. page 1 line 24 a different spelling of the Icelandic volcano is used... Remarks accepted. The article will be re-edited and English will be improved in comparison with the first version. The name of the volcano in Iceland has been corrected.

2. A lot of the references are quite old, the thesis was published in 2015, but this paper is 2019, mercury science has moved on... Remarks accepted. The main task in choosing literature was to show the current state of research in the field of volcanology. The authors provided links to articles where the main study was volcanic activity and

mercury emissions during the eruption period.

3. This paper demonstrates that some of the peaks in the long-term trends of measurements could come from volcanic eruptions that occurred in 2010 and 2011 in Iceland. There are similar sized peaks in 2012 and 2013 that are not identified or discussed that probably aren't volcanic eruptions. What are their sources? Remarks accepted. Indeed, on the presented graphs it can be seen that there are high values of mercury concentration in the spring period. Identification of these the registration of elevated atmospheric mercury concentrations is of significant complexity. Atmospheric transport is the main source of mercury to the polar regions from southern and middle latitudes. Therefore, it is not possible to correct assessment the source of mercury intake. Thus, in this article, the authors showed the real possibility of the correlation of high concentrations of atmospheric mercury with volcanic eruptions in Iceland.

4. The authors should discuss the whole data set and identify the other peaks, or they should not include them and talk about in more detail the volcanic data. Remarks accepted. Taking into account the received comments, we will edit the results was presented in the article, taking into account the spring high values of atmospheric mercury concentration.

---

## Author Comment (AC1) · 28 Mar 2019

1. Essential information such as description of the site, its climatology, instrument calibration, is missing. Remarks partially accepted. A brief description of the site is sufficient for the purposes of this article. The authors believe that a too detailed description of climatology in the paper is excessive (more detail Pankratov et. al. 2010, page 3: line 2). The article provides the most significant information on the measurement modes. There indicated the accumulation period (30 minutes) and the desorption mode (thermal desorption). More information on calibration can be obtained from the link provided in the article on the Tekran website. 2. such as wrong units for the dis-

cussed trends Remarks accepted. Units have been changed (page 1: line 14-17). 3. Third, I think that episodic transport of air masses with high mercury concentrations from volcanic activity in Iceland would be better discussed in event (and their frequency) terms or as an anomaly of seasonal variation, rather than in terms of trends whose discussion makes a substantial part of the paper. Remarks rejected: Episodic transport of air masses with high mercury concentrations from volcanic activity in Iceland are thoroughly described in section 3.2 "Episodes of volcanic eruptions" as an anomaly of seasonal variation. The calculation of the trend for periods of volcanic eruptions made it possible to estimate the degree of increase in the concentration of atmospheric mercury. This approach suggested that the increase in mercury concentrations in spring was associated with volcanic eruptions in Iceland (page 4: line 28-29, 40, page 5: line 5-6. 4. the most recent estimate of volcanic mercury source by Pirrone et al. (Atmos. Chem. Phys., 10, 5951-5964, 2010) is not mentioned. The remark has been accepted. The reference "Pirrone et al. (Atmos. Chem. Phys., 10, 5951-5964, 2010)" will be added to the list of references (page 10: line 11). 5. Pankratov's own paper in Russian Meteorology and Hydrology (Vol. 38, 405-413, 2013) with some experimental details on the Amderma station is not mentioned either. The remark has been accepted. This reference "Russian Meteorology and Hydrology (Vol. 38, 405-413, 2013)" will be added to the list of references (page 9: line 34). 6. What does it mean "we estimate the long-range transport" – flux or what? Reply to comment. The term "long-range transport" means the transfer of a pollutant with air masses over long distances. 7. New data for volcanic eruptions in Iceland Reply to comment. For the first time, an increase in the concentration of elemental mercury in the surface layer of the atmosphere at a particular point in the Russian Arctic was associated with volcanic eruptions.. 8. A change in dynamics" – dynamics of what – transport, meteorology Reply to comment. In this case, we talk about the dynamics of atmospheric mercury for the period when the air masses transferred a significant amount of mercury over the measurement site of the monitoring from the south-west direction.

9. For seasonal variability . . . a negative trend of.. ng/month was fixed" – a trend of

seasonal variability or of concentrations? Remarks accepted. The mercury concentration was expressed as ng per month (page 7: line 9-10).

10. The inverse trajectories. . ." backward trajectories are probably meant. Remarks accepted (page 6: line 23).

11. The last sentence reads either as if volcanic emissions were dominating source of atmospheric mercury in the northern hemisphere or as if Amderma station were representative for the northern hemisphere or both. Both is wrong. Reply to comment. In this case, the assumption is confirmed that the maximum amount of mercury from volcanic eruptions supply to the environment in a short period of time. This statement is based on data from an ice core study. (Krabbenhoft, D.P. and Schuster, P.F., 2002 ,Glacial Ice Cores Reveal A Record of Natural and Anthropogenic Atmospheric Mercury Deposition for the Last 270 Years: 2002 U.S. Geological Survey Fact Sheet FS-051-02, p. 2.)

12. Introduction: Line 34: "we estimate it at about 800 Mg yr-1" – there is neither a reference nor an estimation presented in this paper. Remarks accepted. Link to article will be added (page 1: line 36).

13. The citation of M. Li Witt, 2010 is a conference paper difficult to obtain. In addition, the initials of Ms Witt are wrong. Please cite M.L.I. Witt et al. (J. Geophys. Res. 113, B06203, doi:10.1029/2007JB005401, 2008; J. Volcanology Geotherm. Res. 178, 636-643, 2008). Remarks accepted. Link will be corrected (page 9: line 9).

14. Lines 38-40: It is not clear what elevated particle concentrations at Zugspitze have to do with mercury or other trace metals? Reply to comment. An example of registration in the atmosphere of elevated concentrations of other particles during the passage of a volcanic cloud above the site of the monitoring.

15. Line 41 and the last line of the same paragraph say the same, one of the sentences is redundant. Remarks accepted. The paragraph will be deleted. 16. Page 2, lines 32-

33: There are 4 citations of work by Pankratov of which three are conference abstracts difficult to obtain and thus useless to most readers. Please cite only Pankratov Thesis with the doi number which is accessible on internet. Remarks accepted. Link to article will be added (page 9: line 27-37).

17. Why is the paper by Pankratov et al. (Russian Meteorology and Hydrology, Vol. 38, 405-413, 2013) not cited? It provides some valuable details about the Amderma measurements and an analysis of data until 2011. Remarks accepted. Link to article will be added (page 9: line 34).

18. Section 2.1: The site has to be described, possible mercury sources in the vicinity enumerated, and precautions to eliminate contaminated data delineated. There are no details about the air sampling (flow rate, position of inlet, tubing, its length and material), maintenance (exchange of gold traps, etc.) and calibration (frequency of calibration of Tekran response to mercury and of air flow rate) of the Tekran instrument. Please add details. Remarks accepted. More information on calibration can be obtained from the link provided in the article on the Tekran website.

19. Page 3, line 15: Backward trajectory arrival altitude of 500 and 3000 m - is this altitude justified? What is the orography and typical meteorology of the station (possible mercury sources in the surrounding, prevailing wind direction, velocity, height of inversion layer in different seasons)? Reply to comment. To calculate the Backward trajectory using different altitude. Including the altitude of 3000 meters is the height of the main transport in the atmosphere, the free troposphere.

20. Section 3.1: In Fig. 1 the authors show the position of one of the three measurements sites – why not the position of other two? What was the position of site 1, 2 and 3, i.e. distance from the sea? By the way the measurements at the three sites were already compared in the Pankratov et al. 2013 paper. What about local contamination: has it been observed and eliminated from the data? Reply to comment. The location of monitoring sites was described in the article Pankratov, F .: The Dynamics

of Atmospheric Mercury in the Russian Arctic, Thesis, November 2015, DOI: 10.13140 / RG.2.1.4255.1767. This information does not represent a value for the fact of registering elevated mercury concentrations at volcanic eruption in Iceland. Local sources of pollution do not significantly affect the resulting values of mercury concentration over the entire observation period (page 4: line 8).

21. Section 3.2: This section is nominally about seasonal variation but it deals to a large extent with trends. Figure 3 is not helpful either. The authors should divide the discussion into a section about trends (if necessary at all) and into a section about seasonal variation after detrending the data. Reply to comment. In this section, seasonal variations are treated as trends for certain time intervals.

22. Page 4, lines 6-7: "The smallest variability of mercury concentrations during AMDEs. . ." – what does it mean? AMDEs mean high variability, the smallest variability would be an absence of AMDEs. Remarks accepted. We indicate out that this behavior is typical only for the winter period (page 4: line 6-7).

23. Page 4, line 8: "dynamics of mercury behaviour" – why "dynamics" and not simply "mercury behaviour"? Remarks accepted. This sentence will be adjusted.

24. Page 4, lines 11-14: Because of the trend it would be more appropriate to compare the authors data in 2010 and 2011 with corresponding annual data from Ny Alesund and Alert stations. The data can be found in the literature or obtained from the operators of the stations. The term "significantly" should be accompanied by a significance test. Reply to comment. If possible, I will request data from other stations.

25. Page 4, lines 18-27: Trends have to have a unit of concentrations per time. The trends presented here in concentration units cannot thus be trends. Were the trends calculated from all data, from monthly averages or monthly medians? Fig. 2 shows that since about 2011 the frequency of the AMDEs and AMEEs is much higher than in preceding years without changing much the decreasing tendency. Reply to comment. Observation period (2001-2010) of measurements a negative trend (-0.35 ng for

period), and calculated averages for all seasonal (page 4: line 26).

26. Then likely to be caused by changing frequency and duration of depletion and enhancement events instead of the trend of background. I think that to discuss the 2011 and 2012 anomaly in terms of trend is thus wrong. It is also questionable whether 3 years are sufficient to calculate a trend. Reply to comment. The construction of a trend for a three-year period is reasonable, since there are no time and quantitative limitation when analyzing any series of values.

27. Page 4, lines 28-33: Trend is given in ng/period – the unit is wrong. Which period? Short term changes are not trends! Reply to comment. (page 4: line 36 -0.66 ng per period, Fig. 4; page 4: line 40 +0.97 ng per month; page 4: line 41 -0.88 ng per month. 28. Section 4: Trend units are wrong. Remarks accepted. Trend units will be corrected (page 4: line 36; page 4: line 40; page 4: line 41)

29. Page 7, line 7: Mercury measurements at Amderma seem to continue until 2016. By including later data, the anomaly of 2010 and 2011 could be perhaps illustrated better. Figure 1: This figure could serve as a schematic illustration of the paper content (such as required by Environmental Science and Technology) but is unsuitable for a paper. The insert with the map of Island and its volcanoes can be found in every atlas and can thus be omitted. An insert with a map of the Amderma site (and the three sampling locations mentioned in the text) and the potential mercury sources in the surroundings would be more useful. Reply to comment. The possibility of mapping the likely sources of mercury contamination will be considered. 30. Figure 2: The data shown in this figure for 2002- 2004 are very different from those shown in Figure 2 of Pankratov et al. (2013) for the same period. Please explain the difference. Reply to comment. The original data is not changed, a variety of mathematical techniques used to construct graphs. 31. Figure 3: The decreasing trend in Figure 2 for 2010 – 2013 is now an increasing trend in Figure 3? This only shows that the discussion of volcanic influence in terms of trends is questionable at best. Reply to comment. For the period 2010-2013 (Fig. 2, page 13: line 4) a downward trend was calculated. The trend to

increase (Fig. 3, page 15: line 4) is calculated for the period 2010-2012. 32. Figure 4: The reader would appreciate the same scale on the x axis – why a) stops at the 22nd week when b) continues until 26th and c) until 25th week? The dotted line in d) is not explained – a running average or what? Reference Konoplev et al. (2005) -the title and the page numbers are wrong. Reply to comment. For the graphs presented in the figure information on the number of weeks is not significant. The link to the article will be corrected (page 9: line 3).

Please also note the supplement to this comment:
https://www.atmos-chem-phys-discuss.net/acp-2018-1228/acp-2018-1228-AC1-supplement.pdf

---

## Author Comment (AC2) · 28 Mar 2019

1. In general, the paper needs English re-editing... There are some spelling problems, e.g. page 1 line 24 a different spelling of the Icelandic volcano is used... Remarks accepted. The article will be re-edited and English will be improved in comparison with the first version. The name of the volcano in Iceland has been corrected (page 1: line 24).

2. A lot of the references are quite old, the thesis was published in 2015, but this paper is 2019, mercury science has moved on... Remarks accepted. The main task in choosing literature was to show the current state of research in the field of volcanology.

The authors provided links to articles where the main study was volcanic activity and mercury emissions during the eruption period.

3. This paper demonstrates that some of the peaks in the long-term trends of measurements could come from volcanic eruptions that occurred in 2010 and 2011 in Iceland.

There are similar sized peaks in 2012 and 2013 that are not identified or discussed that probably aren't volcanic eruptions. What are their sources? Remarks accepted. Indeed, on the presented graphs it can be seen that there are high values of mercury concentration in the spring period. Identification of these the registration of elevated atmospheric mercury concentrations is of significant complexity. Atmospheric transport is the main source of mercury to the polar regions from southern and middle latitudes. Therefore, it is not possible to correct assessment the source of mercury intake. Thus, in this article, the authors showed the real possibility of the correlation of high concentrations of atmospheric mercury with volcanic eruptions in Iceland (page 4: line 8-15).

4. The authors should discuss the whole data set and identify the other peaks, or they should not include them and talk about in more detail the volcanic data.

Remarks accepted. Taking into account the received comments, we will edit the results was presented in the article, taking into account the spring high values of atmospheric mercury concentration (page 4: line 8-15).

Please also note the supplement to this comment:
https://www.atmos-chem-phys-discuss.net/acp-2018-1228/acp-2018-1228-AC2-supplement.pdf
* * *
[Figure]

**Supplement:**

**Elevated atmospheric mercury concentrations at the Russian Polar** station Amderma during Icelandic volcanoes' eruptions**

Fidel Pankratova,\*, Alexander Mahurab, Tuukka Petäjäb, Valentin Popovc, Vladimir Masloboeva

a Institute of Northern Environmental Problems, Kola Science Center, Russian Academy of Sciences, Fersman Str. 14A, 5 Apatity, 184200, Russia.

Correspondence to: Fidel Pankratov (fidel\_ru@mail.ru)

- 10 Abstract. We estimate the long-range atmospheric transport of elemental mercury in the Northern Hemisphere and present new data for volcanic eruptions in Iceland. At the Polar station Amderma (Russia) of long-term observations of elemental mercury concentration (2009-2010), a change in the dynamics was recorded. For seasonal variability at the period from 2001-2009 negative trend (-0.66 ng per month) was fixed. However, the analysis of the last three years of measurement (2010-2012) showed the greatest positive trend (+0.97 ng per month). In April 2010 and the highest positive trend was
- observed (+0.24 ng per month), for the first time for the whole (2001-2013). At the same time, high concentrations of 15 gaseous elemental mercury in the range from 1.81 to 2.58 ng m-3 in Apr-Jun 2010 and from 1.81 to 3.31 ng m-3 in May-Jun 2011 in contrast to the typical concentrations of 1.51 ng m-3. During the period of 2010 and 2011 intensified volcanoes in Iceland and consequently volcanic eruptions in Iceland were considered the most probable cause of these increased concentrations. Until now, there have been no cases of recording a high concentration of mercury during the active eruption
- 20 of the volcano, measured so far from the source of the eruption. In this way for the first time at the Amderma station in the Russian Arctic, high levels of elemental mercury were recorded as associated with the periods of active volcanoes Eyjafjallajökull (in 2010) and Grimsvotn (in 2011). The inverse trajectories calculated for a vertical profile covering a height of 500 and 3000 m above sea level the time level with high mercury concentrations confirmed that this was due to atmospheric transport from the northwest and was associated with the active Eyjafjallajökull and Grimsvotn volcanoes.
- 25 Therefore, it can be assumed that these active volcanoes are the main sources of increased mercury concentrations in the northern hemisphere as a result of atmospheric transport of volcanic clouds to the monitoring point in the Russian Arctic.

**1. Introduction**

30

estimated relative to other compounds that are formed during degassing and eruption. During a volcanic eruption about 45 different trace elements from the Earth's crust can enter the atmosphere. However, there is no yet direct correlation between the amounts of the substance emitted into the atmosphere during an eruption and the number of reported cases of volcanic eruptions (Mambo et al., 2001). Volcanoes are considered to be the main natural sources of mercury entered into the atmosphere (Krabbenhoft et al., 2002). The volcanogenic Hg flux from passively degassing volcanoes is about 30 Mg Hg yr

Many studies of global volcanogenic Hg emissions show that, in the main, the arrival of atmospheric mercury can be

- 35 1. The flux from erupting volcanoes is much larger; we estimate it at about 800 Mg yr-1 and geothermal sources contribute to the atmosphere roughly 60 Mg yr-1 Hg (Varekamp et al., 1986). Therefore, current estimate of global mercury emissions suggests that the overall contribution from natural sources and anthropogenic sources is nearly 7527 Mg yr-1 (Pirrone 
[revised manuscript text omitted]

During the whole monitoring period, high values of mercury concentrations in the surface layer of the atmosphere were also recorded. The monitoring point is located at a long distance from regional industrial centers. Given the fact that there are no

- 10 significant sources of mercury emissions at the Amderma polar station, based on calculated trajectories have identified region for industrial enterprises. This approach allows us to estimate the detection of the main emission sources in the Polar region, as well as in middle latitudes. At the same time, it is necessary to take into account the role of the mechanisms of transformation of substances in the process of atmospheric transport. In the case of volcanic eruptions in Iceland (2010– 2011), it became possible to compare cases with increasing mercury concentrations in the atmospheric surface layer and
- 15 periods of volcanic eruptions.

[revised manuscript text omitted]

Witt M. L.I., Pyle D.M., Mather T.A., Aiuppa A., Bagnato E., R.S. Martin R.S.: The Importance of Volcanoes and

15 Geothermal Sources of Mercury to the Atmosphere, 5th International Conference on Heavy Metals in the Environment; Sources, emissions and control of heavy metals, 2010, 924–928.

11